# Low Levels of IgM and IgA Recognizing Acetylated C1-Inhibitor Peptides Are Associated with Systemic Lupus Erythematosus in Taiwanese Women

**DOI:** 10.3390/molecules24091645

**Published:** 2019-04-26

**Authors:** Kai-Leun Tsai, Chen-Chung Liao, Yu-Sheng Chang, Ching-Wen Huang, Yu-Chu Huang, Jin-Hua Chen, Sheng-Hong Lin, Chih-Chun Tai, Yi-Fang Lin, Ching-Yu Lin

**Affiliations:** 1Division of Allergy, Immunology, and Rheumatology, Department of Internal Medicine, Shuang Ho Hospital, Taipei Medical University, New Taipei City 23561, Taiwan; kelen109@hotmail.com (K.-L.T.); risea65@gmail.com (Y.-S.C.); koalalin@gmail.com (S.-H.L.); 2Division of Allergy, Immunology, and Rheumatology, Department of Internal Medicine, School of Medicine, College of Medicine, Taipei Medical University, Taipei 11031, Taiwan; 3Proteomics Research Center, National Yang-Ming University, Taipei 112, Taiwan; ccliao@ym.edu.tw; 4School of Medical Laboratory Science and Biotechnology, College of Medical Science and Technology, Taipei Medical University, Taipei 11031, Taiwan; G660104012@tmu.edu.tw (C.-W.H.); m609105005@tmu.edu.tw (Y.-C.H.); 5Graduate Institute of Data Science, College of Management, Taipei Medical University, Taipei 11031, Taiwan; jh_chen@tmu.edu.tw; 6Research Center of Biostatistics, College of Management, Taipei Medical University, Taipei 11031, Taiwan; 7Department of Laboratory Medicine, Taipei Medical University-Shuang-Ho Hospital, Taipei Medical University, New Taipei City 23561, Taiwan; 08046@s.tmu.edu.tw (C.-C.T.); 08310@s.tmu.edu.tw (Y.-F.L.); 8Department of Biotechnology and Animal Science, National Ilan University, Ilan 26047, Taiwan; 9PhD Program in Medical Biotechnology, College of Medical Science and Technology, Taipei Medical University, Taipei 11031, Taiwan

**Keywords:** systemic lupus erythematosus, C1-inhibitor, acetylation, autoantibody isotype, serum

## Abstract

The objective of this study was to identify novel acetylation (Ac) modifications of the C1-inhibitor (C1-INH) and explain the association of the levels of autoantibodies against acetylated C1-INH peptides with the risk of developing systemic lupus erythematosus (SLE). Ac modifications of the C1-INH were identified and validated through in-gel digestion, nano-liquid chromatography-tandem mass spectrometry, immunoprecipitation, and Western blotting by using serum protein samples obtained from patients with SLE and age-matched healthy controls (HCs). In addition, the levels of serum C1-INH, Ac-protein adducts, and autoantibodies against unmodified and acetylated C1-INH peptides were measured. C1-INH levels in patients with SLE were significantly lower than those in HCs by 1.53-fold (*p* = 0.0008); however, Ac-protein adduct concentrations in patients with SLE were significantly higher than those in HCs by 1.35-fold (*p* = 0.0009). Moreover, immunoglobulin M (IgM) anti-C1-INH^367–385^ Ac and IgA anti-C1-INH^367–385^ Ac levels in patients with SLE were significantly lower than those in HCs. The low levels of IgM anti-C1-INH^367–385^ (odds ratio [OR] = 4.725, *p* < 0.001), IgM anti-C1-INH^367–385^ Ac (OR = 4.089, *p* = 0.001), and IgA anti-C1-INH^367–385^ Ac (OR = 5.566, *p* < 0.001) indicated increased risks for the development of SLE compared with HCs.

## 1. Introduction

Systemic lupus erythematosus (SLE) is a multisystem, chronic autoimmune disease characterized by a variety of variable clinical manifestations and a heterogeneous group of pathogenic autoantibodies produced through a breakdown of tolerance to nucleic acids and proteins, especially chromatin [1,2]. According to the Taiwanese National Health Insurance Research Database, between 2003 and 2008, the average prevalence of SLE in Taiwan was 97.5 new cases (female-to-male ratio, 7.8) per 100,000 persons observed for 1 year; moreover, the highest prevalence in women was observed among those aged 30–39 years, and that in men was observed among those aged 70–79 years. The average SLE incidence rate was 4.87 new cases (female-to-male ratio, 7.0) per 100,000 person-years; the highest incidence rate in women was observed among those aged 40–49 years, and that in men was observed among those aged >70 years. The average standardized mortality rate from SLE was 11.1 new cases (female-to-male ratio, 4.5) per 100,000 person-years [1]. The etiology and pathogenesis of SLE include genetic ancestry, environmental exposure, amines, viruses, post-translational modifications (PTMs), autoantibodies, and periodontitis [2,3,4,5,6].

Research reported several autoantibodies for detecting SLE, including antinuclear antibodies, anti-double-stranded DNA antibodies, anti-Smith antibodies, antinucleosome antibodies, antihistone antibodies, antiribosomal P antibodies, antiphospholipid antibodies, anticomplement component 1q antibodies, antiribonucleoprotein antibodies, and antiproliferating cell nuclear antigen antibodies [7]. Other autoantibodies against specifically modified epitopes from biological fluids are recognized in SLE, including peroxynitrite (H1 histone), 4-hydroxy-2-nonenal (H2A histone), malondialdehyde (MDA; catalase, superoxide dismutase, and human epithelial type 2 protein), trimethylation (H3 histone), isomerization (H2A histone), carbamylation (fetal calf serum proteins), nitration (poly L-tyrosine), and acetylation (Ac; H4, H2A, and H2B histones) [8,9,10,11,12,13,14,15,16].

Several case reports have indicated that patients with SLE exhibited a C1-inhibitor (C1-INH) deficiency that caused serum C1-INH depletion owing to protein dysfunction [17,18,19]. The C1-INH, a plasma protease C1-INH, is a serine-type endopeptidase inhibitor that controls C1 complex activation and plays critical roles in regulating essential pathways including activation of the human complement system, fibrinolysis, blood coagulation, and kinin system generation [18,20]. Furthermore, Meszaros et al. reported that the level of anti-C1-INH immunoglobulin G (IgG) in patients with an SLE-acquired C1-INH deficiency was significantly higher than that in healthy controls (HCs) [21]. Mandle et al. proposed that autoantibody against the reactive center region of C1-INH results in acquired C1-INH deficiency [22].

In the present study, C1-INH modification was identified through one-dimensional sodium dodecylsulfate-polyacrylamide gel electrophoresis (1D SDS-PAGE), in-gel digestion, and label-free nano-liquid chromatography-tandem mass spectrometry (nano-LC-MS/MS) by using serum proteins obtained from patients with SLE versus HCs. Moreover, modifications of the C1-INH were confirmed using immunoprecipitation (IP) and Western blotting. Serum levels of the C1-INH, modified protein adducts, and autoantibodies were determined. Subsequently, we evaluated the associations of the C1-INH, modified protein adduct, and autoantibody levels with risks of SLE pathogenesis in patients with SLE versus HCs.

## 2. Results

### 2.1. Identification and Validation of Novel Ac Modifications of Serum C1-INH

Novel Ac modifications of the C1-INH in serum were identified from a single pair of each of the nine pooled serum samples (HCs vs. patients with SLE) by using 1D SDS-PAGE, in-gel digestion, nano-LC-MS/MS, and PTM finder in-house program through manual examination of modified spectra (Figure 1A,B and Appendix A). The average coverage of amino acid sequences in the C1-INH was 32% (Appendix A). The acquired MS/MS spectra of acetylated peptides in the C1-INH are presented in Figure 1C,D. The peptide moiety was identified as b- and y-series ions. The peptide ^367^-LEDMEQALSPSVFKAIMEK-^385^ was identified as SLE specific and had an Ac modification at lysine (K) 380 with a mass increase of 42.010567 Da. The peptide modified at K380 was identified as an unmodified b14 ion followed by a modified y6 ion (Figure 1C). The initial masses of ^367^-LEDMEQALSPSVFKAIMEK-^385^ at charge states of 1 (*z* = 1) and 3 (*z* = 3) were 2165.075 and 722.699 Da, respectively. The masses of ^367^-LEDMEQALSPSVFKAIM*EK-^385^ charge states of 1 (*z* = 1) and 3 (*z* = 3) were 2226.108 and 742.036 Da, respectively (Figure 1B). The peptide ^310^-MEPFHFKNSVIKVPMMNSK-^328^ was determined to be HC-specific. An Ac modification with a mass increase of 42.010567 Da was identified at K316 and K321. The peptides modified at K316 and K321 were presented as an unmodified b7 ion followed by a modified y13 ion and unmodified b12 ion followed by a modified y8 ion, respectively (Figure 1D). The initial masses of ^310^-MEPFHFKNSVIKVPMMNSK-^328^ at charge states of 1 (*z* = 1) and 3 (*z* = 3) were 2263.132 and 755.384 Da, respectively. The masses of ^310^-M*EPFHFKNSVIKVPM*MNSK-^328^ at charge states of 1 (*z* = 1) and 3 (*z* = 3) were 2382.174 and 794.058 Da, respectively (Figure 1B).

Novel Ac modifications of the C1-INH in serum were validated using IP–Western blotting (Figure 2). The Ac modifications of the C1-INH were confirmed in a pair of individual or pooled serum samples (20 pairs of HCs vs. patients with SLE) through IP–Western blotting, which showed a molecular weight of 96–105 kDa (Figure 2A). In the pair of individual serum samples, increased acetylated C1-INH levels were observed in samples obtained from patients with SLE compared with samples from HCs; however, in the pair of pooled serum samples, no difference in C1-INH levels was observed between the samples from the patients and HCs (Figure 2A). Further, the results of IP–Western blotting revealed no difference in C1-INH levels from 20 pairs of individual serum that derived from pooled serum samples (Figure 2B).

### 2.2. Determination of Serum C1-INH Levels Using Western Blotting

Serum protein levels of the C1-INH were determined through Western blotting. The results revealed that C1-INH levels in patients with SLE were significantly lower than those in HCs by 1.53-fold (*p* = 0.0008; Figure 3A). Equal amounts of serum proteins were observed in this experiment (Figure 3A, right bottom panel). The area under the receiver operating characteristic (ROC) curve (AUC) value, sensitivity, and specificity of the serum C1-INH levels in patients with SLE versus HCs were estimated on the basis of the ROC curve. The results obtained from Western blotting indicated that the AUC value was 0.73, sensitivity was 77.5%, and specificity was 52.5% for SLE measurement at an optimal cutoff value of 255624.4 (Figure 3B).

### 2.3. Autoantibodies Against C1-INH^367–385^ and C1-INH^367–385^ Ac Peptides

Autoantibody isotypes recognizing the C1-INH^367–385^ and C1-INH^367–385^ Ac peptides were evaluated using an enzyme-linked immunosorbent assay (ELISA). The significance level of the executed one-way analysis of variance (ANOVA) was set to *p* < 0.0167. The antibody titer of IgM anti-C1-INH^367–385^ in patients with rheumatoid arthritis (RA) was significantly higher than that in patients with SLE by 1.41-fold (*p* = 0.0056); however, the IgM anti-C1-INH^367–385^ Ac titer in patients with SLE was significantly lower than that in HCs by 1.40-fold (*p* = 0.0095; Figure 4, left panel). The levels of IgG anti-C1-INH^367–385^ and IgG anti-C1-INH^367–385^ Ac did not significantly differ among patients with SLE, patients with RA, and HCs (Figure 4, middle panel). The antibody titer of IgA anti-C1-INH^367–385^ Ac in patients with SLE was significantly lower than that in HCs by 1.36-fold (*p* = 0.0004), and that in patients with RA was significantly higher than that in patients with SLE by 1.31-fold (*p* = 0.0012). However, the levels of IgA anti-C1-INH^367–385^ did not significantly differ among patients with SLE, patients with RA, and HCs (Figure 4, right panel).

### 2.4. Determination of Serum Ac-Protein Adducts

Serum concentrations of the Ac-protein adducts in patients with SLE were significantly higher than those in HCs by 1.35-fold (*p* = 0.0009; Appendix A, upper panel). The results of the ELISA conducted for determining the serum Ac-protein adduct revealed that the AUC value was 0.67, sensitivity was 70.4%, and specificity was 50.0% for SLE determination at an optimal cutoff value of 0.299 (Appendix A, bottom panel).

### 2.5. Associations of Decreased C1-INH Levels, Elevated Ac-Protein Adduct Levels, and Reduced Autoantibody Titers against C1-INH^367–385^ and C1-INH^367–385^ Ac Peptides with Risks in Patients with SLE

Reduced levels of IgM anti-C1-INH^367–385^, IgM anti-C1-INH^367–385^ Ac, and IgA anti-C1-INH^367–385^ Ac were associated with 4.725-fold (*p* < 0.001, power = 0.737), 4.089-fold (*p* = 0.001, power = 0.892), and 5.566-fold (*p* < 0.001, power = 0.848) higher risks, respectively, for the development of SLE compared with HCs, indicating a significant difference after adjustment for age in the logistic regression analysis (Table 1). In cases where the power value was <0.7 or the risk for development of SLE did not differ significantly from that of HCs, the results of the age-adjusted odds ratios (ORs) were not considered for the levels of the following: The C1-INH, Ac-protein adduct, IgG anti-C1-INH^367–385^, IgA anti-C1-INH^367–385^, and IgG anti-C1-INH^367–385^ Ac (Table 1).

## 3. Discussion

This is the first study to identify novel Ac modifications of serum C1-INH and to compare the autoreactivity against acetylated C1-INH peptides of patients with SLE with that of HCs. Protein Ac and deacetylation, N-terminal Ac, and lysine Ac are reversible and pivotal for many vital cellular processes [23]. Histone protein Ac can stimulate gene expression and perform chromatin remodeling and transcriptional activation through destabilizing histone–histone and histone–DNA interactions [24]. Spange et al. reported that acetylated non-histone proteins influenced signaling, DNA binding, protein–protein interaction, localization, and the degradation and function of proteins; furthermore, acetylated non-histone proteins were related to immune functions and tumorigenesis [25]. An imbalance between lysine acetyltransferases and lysine deacetylases can cause various diseases, including autoimmunity, diabetes, neurodegenerative disorders, cardiac hypertrophy, and cancer [23]. In our study, we identified and verified novel Ac modifications (K316, K321, and K380) of serum C1-INH (Figure 1, Figure 2 and Appendix A); additionally, the results obtained from the observation of autoantibody isotypes against unmodified and acetylated C1-INH peptides from patients with SLE and HCs are presented in Figure 4.

The C1-INH is an acute-phase protein belonging to the serpin superfamily, and the protein’s level rises during inflammation [26]. The C1-INH can control proteases, including kinin (plasma kallikrein), complements (C1r and C1s), plasmin (fibrinolysis), and coagulation factors (Xla, Xlla, and XIIf) involved in inflammatory responses [27]. The C-terminal serpin domain of the C1-INH may exhibit inhibitory activity [28]. The N-terminal domain of the C1-INH may display anti-inflammatory properties in diseases other than hereditary angioedema (HAE) [29]. The native form of the C1-INH can inhibit plasma kallikrein, the C1 complex, and plasmin; however, plasmin can also degrade the denatured form of the C1-INH [28]. Ansari et al. proposed that lysine Ac causes destabilization of the native protein conformation structure [30]. Dhillon and Adams reported that patients with SLE experience abnormalities in fibrinolysis (e.g., impaired fibrinolysis) [31]. Plasmin proteolytic cleavage of the C1-INH was reported to cause the loss of protease inhibition that occurs during inflammatory processes [27]. In this study, patients with SLE exhibited low C1-INH levels, and the C1-INH levels in patients with SLE were significantly lower than those in HCs, whose levels were on average 1.53-fold higher than those of patients with SLE (Figure 3A). The decreased levels of C1-INH (AUC = 0.73) demonstrated acceptable discriminative value for distinguishing patients with SLE from those without (Figure 3B).

In our study, Ac-protein adduct levels were significantly higher in patients with SLE compared with HCs (Appendix A). A report from van Bavel et al. revealed that apoptosis-associated Ac on histone H2B was an epitope resulting in autoantibody production in a prediseased lupus mouse model [32]. Also using a lupus mouse model, Dieker et al. indicated that apoptosis-induced Ac of histone 4 (H4) may play a pathogenic role [16]. Another study proved that autoantibodies against acetylated histone peptides (H2B and H4) were correlated with disease activity in SLE [12]. Thus, Ac-protein adducts can induce autoreactivity to increase the pathogenic risk in patients with SLE; this also signifies that autoantibodies can neutralize Ac-protein adducts. Furthermore, Alaskhar Alhamwe et al. reported that histone modifications including acetylation, phosphorylation, methylation, and ubiquitination may play regulating roles in the development of allergic diseases [33]. In addition to histone acetylation, we investigated other histone modifications that may play a role in the development of SLE. Lupus-derived antibody LG11-2 can react with H2BK14 after apoptosis-induced phosphorylation [32,34]. Additionally, van Bavel et al. found that apoptosis-induced methylation of H3K27 was targeted by autoantibodies in SLE [14]. Suzuki et al. revealed that antihistone antibodies can bind to ubiquitinated H2A in SLE [35]. Additionally, histone modifications can also affect the development of RA [36]. Lloyd et al. indicated that autoantibody reactions with acetylated histone 2B may play a role in RA pathogenesis [37]. Otherwise, studies on autoantibodies targeted by other histone modifications (including phosphorylation, methylation, and ubiquitination) in RA have not been conducted.

Immune system disorders featuring a C1-INH deficiency, including HAE and SLE, involve autoantibodies against the C1-INH [21,38]. Alsenz et al. proposed that anti-C1-INH autoantibodies inactivate the 105-kDa C1-INH to release cleaved 96-kDa C1-INH and thus activate C1s [39]. He et al. revealed that two amino acid sequences, ^446^-LLVF-^449^ and ^452^-QQPF-^455^, may be potential epitopes in the C1-INH that recognize anti-C1-INH autoantibodies [40]. Meszaros et al. reported that the anti-C1-INH IgG titer was significantly higher in patients with SLE than in HCs, and an elevated anti-C1-INH IgG titer was correlated with the duration and activity of the disease [21]. Mandle et al. proposed that anti-C1-INH IgG blocks C1-INH inhibition of C1s to activate the complement system [22]. According to our findings, the C1-INH^367–385^ peptide is not an autoantigen for IgM, IgG, or IgA induction. However, the levels of IgM anti-C1-INH^367–385^ and IgG anti-C1-INH^367–385^ in patients with SLE were not significantly lower than those in HCs according to a one-way ANOVA (Figure 4A). Furthermore, the levels of the IgG anti-C1-INH^367–385^ Ac peptide could not be induced in patients with SLE; however, the levels of IgM anti-C1-INH^367–385^ Ac and IgA anti-C1-INH^367–385^ Ac in patients with SLE were significantly lower than those in HCs (Figure 4B). Serum IgM and IgG inductions represent short-term and long-term immune responses that can neutralize autoantigens. Observations of immune responses from short- to long-term, the levels of autoantibodies recognizing C1-INH^367–385^ and C1-INH^367–385^ Ac peptides increased from IgM (0.75-fold and 0.71-fold) to IgG (0.91-fold and 0.94-fold) in patients with SLE compared with HCs (Figure 4, left and middle panels). Thus, Ac-protein adducts could not be removed through short-term immune responses, but the levels of anti-C1-INH^367–385^ Ac autoantibody increased progressively to neutralize the C1-INH^367–385^ Ac peptide in long-term immune responses. Moreover, low levels of IgA anti-C1-INH^367–385^ Ac could not neutralize the C1-INH^367–385^ Ac peptide (Figure 4, right panel). Decreased levels of IgM anti-C1-INH^367–385^, IgM anti-C1-INH^367–385^ Ac, and IgA anti-C1-INH^367–385^ Ac were also associated with increased risk for the development of SLE compared with HCs (Table 1). Hodkinson et al. indicated that patients with low levels of IgA and IgM may be at an increased risk of infection complications at mucosal sites [41]. Moreover, Traverso et al. indicated that anti-MDA peptide autoantibodies can eliminate the accumulation of harmful MDA-modified protein adducts; however, higher serum levels of MDA-modified protein adducts may indicate the presence of harmful proteins that cannot be efficiently removed by autoantibodies [42]. No studies have reported whether elevated Ac-protein adduct levels in patients with SLE are harmful. As suggested by the current study, increased levels of Ac-protein adduct may be a risk factor for SLE; however, the corresponding power was <0.7 (Table 1). Furthermore, autoantibodies against other epitope modifications, including those of *N*-homocysteinylation, citrullination, MDA, malondialdehyde-acetaldehyde (MAA), carbamylation, acetylation and nitration in non-histone proteins have also been implicated in the etiopathogenesis of SLE and RA [10,13,15,43,44,45,46,47]. Especially, anti-citrullinated and anti-acetylated protein antibody response increase the risk of disease relapse in patients with RA following disease modifying antirheumatic drug (DMARD) treatment [48]. In this study, the levels of anti-C1-INH^367–385^ Ac peptide antibodies in compared patients with RA with HCs were not significantly different (Figure 4B). Further, we need identify new Ac modifications of serum C1-INH in patients with RA and measure the levels of anti-acetylated CI-INH peptide antibody in patients with RA compared with HCs.

## 4. Materials and Methods

### 4.1. Patients and Controls

Serum specimens from 144 female patients (54 with SLE (41.17 ± 11.65 years), 40 with RA (54.85 ± 10.66 years), and 50 HCs (42.00 ± 8.41 years)) were collected from the Department of Laboratory Medicine and the Division of Allergy, Immunology, and Rheumatology, Department of Internal Medicine, Shuang-Ho Hospital (New Taipei City, Taiwan). Patients with SLE or RA were diagnosed by rheumatologists and met appropriate classification criteria. Patients with RA met the appropriate classification criteria: Either the 2010 American College of Rheumatology (ACR)/European League Against Rheumatism classification criteria [49] or the 1987 ACR classification criteria [50]. Patients with SLE met the 1997 ACR SLE classification criteria [51]. This study was approved by the Taipei Medical University-Joint Institutional Review Board, and all volunteers signed an informed consent form before participating in the study (No. 201104003 (2011/06/22) and 201501059 (2015/05/09)). The PTM of the serum C1-INH protein was identified in triplicate using 1D SDS-PAGE, in-gel digestion, and nano-LC-MS/MS by using pooled serum protein samples randomly selected from nine patients with SLE and nine age-matched HCs. In addition, the acetylated C1-INH protein was confirmed through IP coupled with Western blotting using serum samples randomly selected from another 20 pairs of pooled or individual serum samples. Serum C1-INH levels were determined through Western blotting using individual serum samples of 40 patients with SLE and 40 HCs. Serum Ac-protein adduct levels were assessed in 54 patients with SLE and 50 HCs. Autoantibodies against unmodified peptides and their acetylated peptides were evaluated among 54 patients with SLE, 40 patients with RA, and 50 HCs (Appendix A). Serum was stored at −20 °C before use. The clinical and demographic characteristics of patients with SLE, patients with RA, and HCs are presented in Appendix A. The ages of patients with SLE did not differ significantly from those of HCs; however, the ages of patients with SLE differed significantly from those of patients with RA (Appendix A).

### 4.2. In-Gel Digestion and PTM Identification Using Nano-LC-MS/MS

Serum protein concentrations were measured using a Bradford protein assay according to the protocol of Chang et al. [52]. In the measurement, 50 µg of pooled serum protein samples from nine HCs and nine patients with SLE was analyzed using 8% SDS-PAGE, and the gel band was cut corresponded to the molecular weights of 96–105 kDa **(**Figure 1A). The in-gel digestion was operated in triplicate according to the protocol of Uen et al. [53]. Tryptic peptides were separated in a nano-flow high-performance liquid chromatography system (Agilent Technologies 1200 series, Waldbronn, Germany) in triplicate by using an Agilent C18 column (100 mm × 0.075 mm, 3.5 μm in diameter) that was coupled with an LTQ-Orbitrap Discovery^TM^ hybrid mass spectrometer with a nanoelectrospray ionization source (Thermo Electron, Waltham, MA, USA). An MS/MS dataset was analyzed using Xcalibur 2.0 SR1 software (Thermo Electron), and peptide sequences were identified using the SEQUEST algorithm against a human protein sequence database (UniProt; http://www.uniprot.org/, 2016/11) [54]. Furthermore, our PTM finder-in-house program was used to identify modified peptide sequences and sites of serum C1-INH [53]. All modified MS_1_ spectra were manually confirmed, and fragmented ions were labeled as b, y, y-NH_3_, and b-H_2_O ions. Detailed methods are presented in Appendix A.

### 4.3. IP and Western Blotting 

The IP of serum C1-INH was performed using a mouse anti-C1-INH monoclonal antibody (M01, Abnova, New Taipei City, Taiwan). Immunoprecipitated C1-INH (100 μg of protein in 8% gel) or serum protein (2 μg of protein in 6% gel) was separated in SDS-PAGE gels and was assessed using Western blotting. Ac modifications of the C1-INH were verified using a rabbit polyclonal acetylated-lysine antibody (#9441 Cell Signaling Technology, Danvers, MA, USA). C1-INH protein levels were evaluated using a mouse anti-C1-INH monoclonal antibody (M01, Abnova). Detailed methods are presented in Appendix A.

### 4.4. Measurement of Autoantibodies against C1-INH^367–385^ and C1-INH^367–385^ Ac Peptides

The 367–385 amino acid sequences of the human C1-INH peptide, namely LEDMEQALSPSVFKAIMEK (named C1-INH^367–385^), and acetylated C1-INH peptide, namely LEDMEQALSPSVFK(Ac)AIMEK (named C1-INH^367–385^ Ac), were synthesized (Yao-Hong Biotechnology, New Taipei City, Taiwan). Autoantibody isotypes (IgG, IgM, and IgA) against the C1-INH^367–385^ and C1-INH^367–385^ Ac peptides were detected using an ELISA according to the protocol of Chang et al. [52]. In total, 144 serum samples were tested in duplicate. The optical density was estimated at 450 nm with the reference standard at 620 nm. Detailed methods are presented in Appendix A.

### 4.5. Determination of Serum Ac-Protein Adducts

Ac-protein adduct concentrations were determined using the ELISA with minor modification according to the protocol of Chang et al. [52]. A total of 104 serum samples were determined in duplicate. Detailed methods are presented in Appendix A.

### 4.6. Statistical Analyses

Student’s *t* test was used to test the significance of differences in Ac-protein adducts and blot densitometry of serum C1-INH levels. Levels of autoantibodies against the C1-INH^367–385^ and C1-INH^367–385^ Ac peptides were tested using a one-way ANOVA among multiple groups. The mean difference between any two groups was determined using Scheffe’s post hoc test, and the significance level was tested using the Bonferroni method with adjusted *p* values (*p* < 0.0167). Data are presented as the mean ± standard deviation. The adjusted OR and their 95% confidence intervals (CIs) for SLE risk were calculated using univariate and multiple logistic regression models, and the corresponding statistical power was estimated. The diagnostic performance, including the AUC value, sensitivity, and specificity, was assessed using ROC curves, and the 95% CIs were calculated. A *p* value of <0.05 was set as the significance level, unless otherwise indicated. Statistical analyses were performed using SAS (version 9.3, SAS Institute, Cary, NC, USA) and GraphPad Prism (version 5.0, Graphpad Software, San Diego, CA, USA).

## 5. Conclusions

We directly identified novel Ac modifications of the C1-INH protein in serum and determined the association of autoantibodies against C1-INH^367–385^ and C1-INH^367–385^ Ac peptides with the risk of SLE development. Our findings indicate that low IgM anti-C1-INH^367–385^, IgM anti-C1-INH^367–385^ Ac, and IgA anti-C1-INH^367–385^ Ac levels are associated with increased risks of the development of SLE. A larger cohort is required to verify the present results.

## Figures and Tables

**Figure 1 molecules-24-01645-f001:**
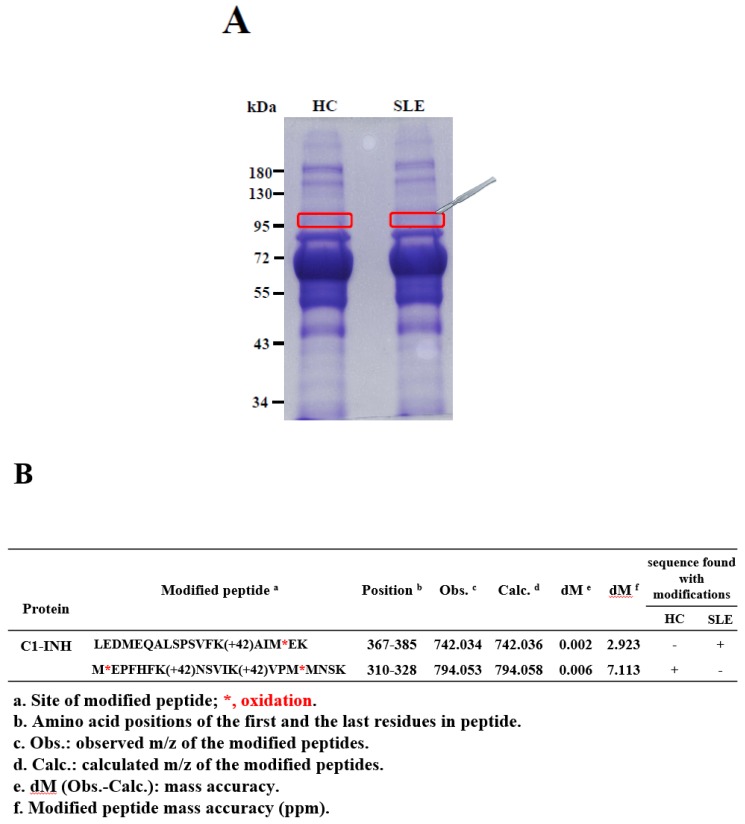
Gel stained with Coomassie Brilliant Blue (CBB) and cut according to molecular weights of 96–105 kDa (**A**). Identification of novel types of acetylation (Ac) modifications of the C1-inhibitor (INH) (**B**). Representative tandem mass spectrometry (MS/MS) spectra of the ^367^-LEDMEQALSPSVFKAIMEK-^385^ peptide sequence and the modified peptide bearing the acetylated K380 residue (**C**). MS/MS spectrum of ^310^-MEPFHFKNSVIKVPMMNSK-^328^ and the modified peptide bearing the Ac-modified sites of K316 and K321 residues (**D**).

**Figure 2 molecules-24-01645-f002:**
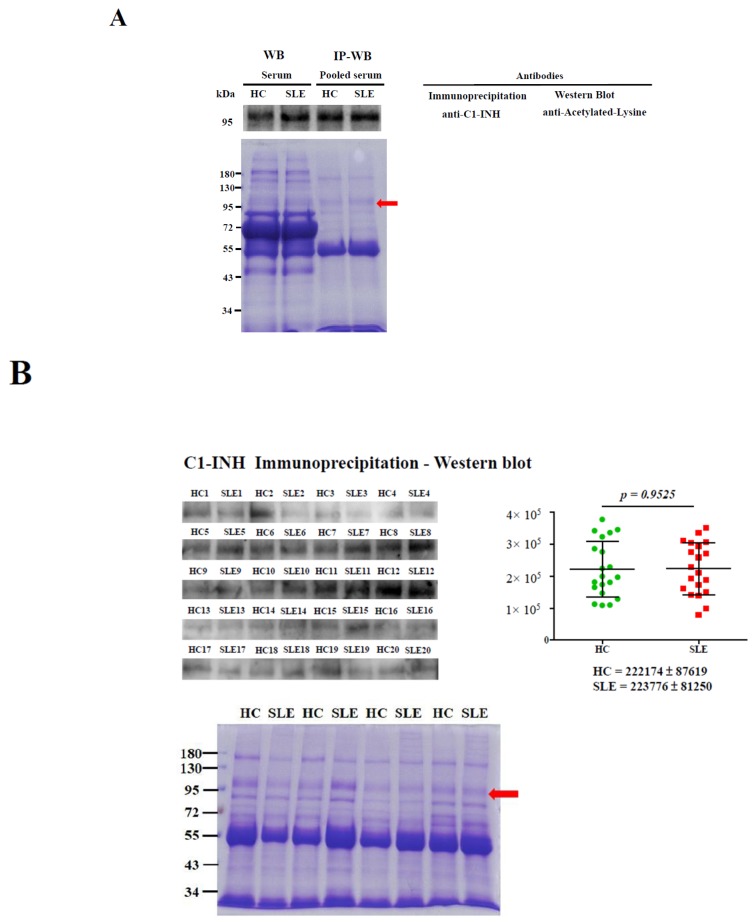
Acetylation modification of the C1-inhibitor (INH) validated using immunoprecipitation (IP) and Western blotting. The percentage of sodium dodecylsulfate polyacrylamide gel electrophoresis (SDS-PAGE) gel was 8%, and IP loading amount of serum proteins was 100 µg of IgG-removal serum proteins. The C1-INH was immunoprecipitated from pooled serum samples (20 healthy controls (HCs) and 20 patients with systemic lupus erythematosus (SLE)) using an anti-C1-INH antibody, and samples were then subjected to Western blotting with an anti-acetylated-lysine antibody (upper panel). Individually selected random serum samples (2 μg protein of HCs and patients with SLE) were used as controls; these were simultaneously used for Western blotting with an anti-acetylated-lysine antibody (**A**). IP–Western blotting was conducted using 20 pairs of aforementioned individual serum samples (**B**). A duplicate SDS-PAGE gel was stained with Coomassie Brilliant Blue (CBB) as the loading control. The red arrow indicates the immunoprecipitated C1-INH.

**Figure 3 molecules-24-01645-f003:**
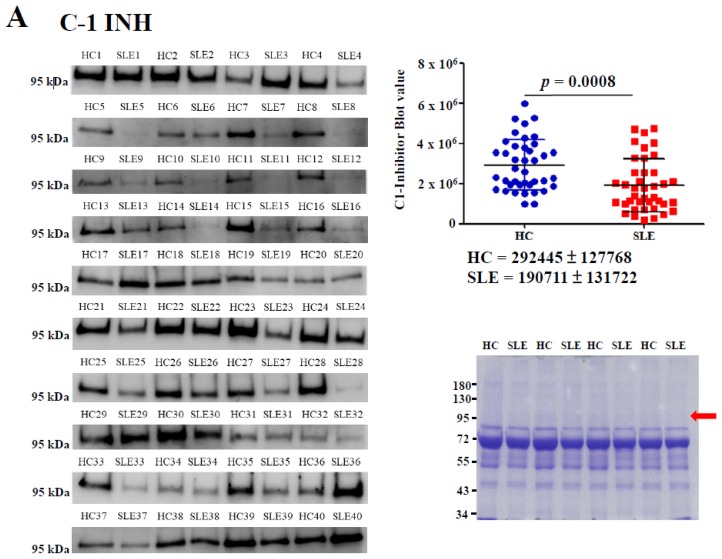
Serum protein levels of the C1-inhibitor (INH) were determined using an anti-C1-INH antibody through Western blotting using 40 pairs of individual serum samples in patients with systemic lupus erythematosus (SLE) versus healthy controls (HCs). Average blot densitometric values were examined using duplicate data (right, upper panel). A 6% sodium dodecylsulfate polyacrylamide gel electrophoresis (SDS-PAGE) gel and 2 μg of serum proteins were used for Western blotting. A duplicate SDS-PAGE gel was stained with Coomassie Brilliant Blue (CBB) as the loading control (right, bottom panel). The red arrow indicates the C1-INH (**A**). Receiver operating characteristic (ROC) curves were plotted according to blot densitometry of the C1-INH. The area under the ROC curve (AUC), sensitivity, and specificity were further calculated (**B**).

**Figure 4 molecules-24-01645-f004:**
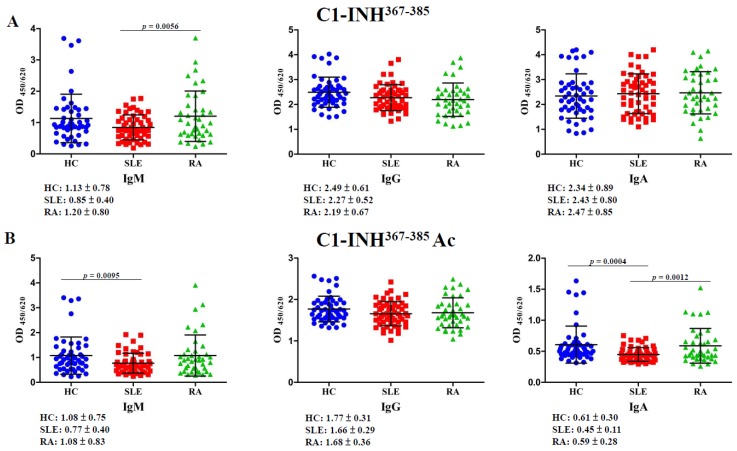
Dot plot of serum concentrations (absorbance units at 450/620 nm) of immunoglobulin M (IgM), IgG, and IgA autoantibody isotypes recognizing the C1-inhibitor (INH)^367–385^ (**A**) and C1-INH^367–385^ acetylation (Ac) (**B**) in 50 healthy controls (HCs), 54 patients with systemic lupus erythematosus (SLE), and 40 patients with rheumatoid arthritis (RA); the plot was obtained using an enzyme-linked immunosorbent assay (ELISA). OD_450/620_, optical density at 450/620 nm.

**Table 1 molecules-24-01645-t001:** Associations of levels of the C1-inhibitor (INH), acetylation (Ac)-protein adduct, and anti-C1-INH^367–385^ and anti-C1-INH^367–385^ Ac peptide autoantibody isotypes with risks of SLE development in 54 patients with systemic lupus erythematosus (SLE) versus 50 healthy controls (HCs).

Risk Factors		Cut-off	HC	SLE	Univariate Logistic Regression Model ^a^			Age-Adjusted Logistic Regression Model		
			N = 50	N = 54	ORs (95% CI)	*p*-Value	Power	ORs (95% CI)	*p*-Value	Power
C1-INH	>	255624.4	21	10	1	0.013	0.697	1	0.015	0.684
	≤	255624.4	19	30	3.315 (1.286, 8.548)			3.250 (1.256, 8.409)		
Acetylation-protein adduct	≤	0.299	25	16	1	0.035	0.568	1	0.035	0.570
	>	0.299	25	38	2.375 (1.062, 5.314)			2.381 (1.063, 5.334)		
IgM anti-C1-INH^367-385^	≥	0.762	38	24	1	0.001	0.663	1	<0.001	0.737
	<	0.762	12	30	3.958 (1.705, 9.189)			4.725 (1.929, 11.573)		
IgG anti-C1-INH^367-385^	≥	2.026	41	33	1	0.021	0.403	1	0.020	0.413
	<	2.026	9	21	2.899 (1.172, 7.169)			2.957 (1.190, 7.349)		
IgA anti-C1-INH^367-385^	<	2.718	37	33	1	0.164	0.231	1	0.183	0.219
	≥	2.718	13	21	1.811 (0.785, 4.178)			1.771 (0.763, 4.108)		
IgM anti-C1-INH^367-385^ Ac	≥	0.855	27	13	1	0.002	0.853	1	0.001	0.892
	<	0.855	23	41	3.702 (1.605, 8.538)			4.089 (1.725, 9.694)		
IgG anti-C1-INH^367-385^ Ac	≥	1.501	43	39	1	0.091	0.250	1	0.090	0.252
	<	1.501	7	15	2.362 (0.872, 6.397)			2.371 (0.874, 6.432)		
IgA anti-C1-INH^367-385^ Ac	≥	0.465	36	17	1	<0.001	0.850	1	<0.001	0.848
	<	0.465	14	37	5.597 (2.409, 13.005)			5.566 (2.380, 13.016)		

^a^ OR, odds ratio.

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
