# Peer review of "Low Levels of IgM and IgA Recognizing Acetylated C1-Inhibitor Peptides Are Associated with Systemic Lupus Erythematosus in Taiwanese Women"

_molecules, 2019, doi:10.3390/molecules24091645_

Round 1
Reviewer 1 Report
Major comments:
1. If I understand it well, the screening for C1-INH-Ac peptides/protein sequences was done only in SLE patients and HC groups. Is it possible that RA patients would reveal any other types of Ac-modified C1-INH peptides/protein sequences specific for RA?
2. If yes, would it be also true for Abs against C1-INH-Ac peptides/protein sequences measured in the second part of the study? Would it be possible that RA patients develop other types of Abs as they produce different types of Ac modifications of C1-INH peptides/protein sequences?
3. Lines 106-108: “In the pair of individual serum samples, increased acetylated C1-INH levels were observed in samples obtained from patients with SLE compared with samples from HCs;”. Where is it shown? In which figure?
4. The first part of this work is a bit unclear, I mean Results 2.1 and 2.2 and accompanying figures/tables. Specifically, it is unclear what, when, in which samples, and with which method was assayed/measured qualitatively/quantitatively. It looks like several things were done several times with different methods and it is unclear what was the final results. It would be great if one could sort it out. Maybe a flow chart would be helpful?
5. Lines 172-176. It is a bit unclear. What you mean “were not accepted”?
6. Antibodies against other PT-modified proteins have also been implicated in the etiopathogenesis of SLE (PMID: 17718048) or RA (PMID: 24015711) or at least associated with those diseases. Please, refer to in your Discussion.
7. Lines 253-256 and Supplementary Table 2. “age-matched”? If yes, only between SLE and HC. How could it affect the data obtained for RA in relation to those obtained for SLE and HC (in addition to major points 1 and 2)? Is it possible that the different age in RA group could mask the real results in this group?
Other comments:
1. Some sentences or their parts are unclear. E.g. lines 37-38: “indicated increased risks of pathogenesis in patients with SLE compared with HCs.” What do you mean “pathogenesis” in this context?
2. Some constructions are repeated in (directly) neighboring sentences. E.g. lines 69-72: “In this study”. Where the paragraph telling what you did in the present study really starts then?
3. Figure 2A. The right panel of this figure (with Abs) is unclear. What it is and what it corresponds to? One can made an educated guess but it is not super clear. Besides, what is the relationship between WB and IP-WB (see also major comment 4)?
4. Line 127: “Equal amounts”. What do you mean? How it corresponds to Figure 3A?
5. Lines 130-132 and Figure 3B. The combination of sensitivity and specificity you gave is the combination for the best/optimal criterion? The same for lines 162-166.
6. Lines 185-187. It is a very nice, synthetic, and informative sentence on the relationship between histone acetylation and gene expression. However, please, add 1-2 sentences on other types of histone PT modifications (PMID: 29796022)? Could these other types of PT histone modifications also play a role in SLE and/or RA?
Author Response
Dear Prof. Dr. Paula A. C. Gomes,
We greatly appreciate your interest in our manuscript, “Low Levels of IgM and IgA Recognizing Acetylated C1-Inhibitor Peptides Are Associated with Systemic Lupus Erythematosus in Taiwanese Women” (molecules-463336). We thank the reviewers for their constructive comments and suggestions. We have revised the manuscript accordingly. All the revisions have been highlighted in red in the revised version.
Our responses to the comments and suggestions of reviewer 1 are as follows:
Q1: If I understand it well, the screening for C1-INH-Ac peptides/protein sequences was done only in SLE patients and HC groups. Is it possible that RA patients would reveal any other types of Ac-modified C1-INH peptides/protein sequences specific for RA?
A1: Yes. It is possible.
Q2: If yes, would it be also true for Abs against C1-INH-Ac peptides/protein sequences measured in the second part of the study? Would it be possible that RA patients develop other types of Abs as they produce different types of Ac modifications of C1-INH peptides/protein sequences?
A2: Yes. It is possible.
Q3. Lines 106-108: “In the pair of individual serum samples, increased acetylated C1-INH levels were observed in samples obtained from patients with SLE compared with samples from HCs;”. Where is it shown? In which figure?
A3: In Figure 2A, blot densitometry of individual serum sample from patient with SLE was observably higher than samples from HCs. See the follow picture (Figure 2A, upper left panel).
Q4: The first part of this work is a bit unclear, I mean Results 2.1 and 2.2 and accompanying figures/tables. Specifically, it is unclear what, when, in which samples, and with which method was assayed/measured qualitatively/quantitatively. It looks like several things were done several times with different methods and it is unclear what was the final results. It would be great if one could sort it out. Maybe a flow chart would be helpful?
A4: We have added a flow chart in Supplementary Figure 1.
Supplementary Figure 1 Flow chart.
Q5: Lines 172-176. It is a bit unclear. What you mean “were not accepted”?
A5: We have revised the text “In cases where the power value was <0.7 or the risk for development of SLE did not differ significantly from that of HCs, the results of the age-adjusted odds ratios (ORs) were not considered for the levels of the following: the C1-INH, Ac-protein adduct, IgG anti-C1-INH367-385, IgA anti-C1-INH367-385, and IgG anti-C1-INH367-385Ac (Table 1).” in lines 177-180 on page 8.
Q6: Antibodies against other PT-modified proteins have also been implicated in the etiopathogenesis of SLE (PMID: 17718048) or RA (PMID: 24015711) or at least associated with those diseases. Please, refer to in your Discussion.
A6: We added the texts “Furthermore, autoantibodies against other epitope modifications, including those of N-homocysteinylation, MDA, carbamylation, and nitration in non-histone proteins have also been implicated in the etiopathogenesis of SLE and RA.” in lines 266-268 on page 10.
Q7: Lines 253-256 and Supplementary Table 2. “age-matched”? If yes, only between SLE and HC. How could it affect the data obtained for RA in relation to those obtained for SLE and HC (in addition to major points 1 and 2)? Is it possible that the different age in RA group could mask the real results in this group?
A7: We have revised the text “50 age-matched HCs” to “50 HCs” in line 272 on page 11. RA is disease control. In this study, we investigate the association of levels of autoantibodies against acetylated C1-INH peptides compared SLEs to HCs. Thus, it is not affected the data obtained for RA in relation to those obtained for SLE and HC.
Q8: Some sentences or their parts are unclear. E.g. lines 37-38: “indicated increased risks of pathogenesis in patients with SLE compared with HCs.” What do you mean “pathogenesis” in this context?
A8: We have revised the text “indicated increased risks for the development of SLE compared with HCs” in lines 37 on page 1. We have revised the text “of developing systemic lupus erythematosus (SLE)” in lines 26 on page 1. We have revised the text “for the development of SLE” in line 175 on page 8. We have revised the text “the risk for development of SLE did not differ significantly from that of HCs” in lines 177-178 on page 8. In the legend of Table 1, we have revised the “SLE development” in line 182 on page 8. We have revised the text “for the development of SLE” in lines 257-258 on page 10. We have revised the text “SLE development” in line 347 on page 12. We have revised the text “of the development of SLE” in line 348 on page 12.
Q9: Some constructions are repeated in (directly) neighboring sentences. E.g. lines 69-72: “In this study”. Where the paragraph telling what you did in the present study really starts then?
A9: We have deleted the texts “In this study, we investigated the association of isotypes of autoantibodies against the novel PTM of C1-INH with risks of pathogenesis in patients with SLE. In this study, we investigated isotypes of autoantibody against the novel modification of C1-INH in Taiwanese women with SLE.” on page 2. We added the text “In the present study, the” in line 72 on page 2.
Q10: Figure 2A. The right panel of this figure (with Abs) is unclear. What it is and what it corresponds to? One can made an educated guess but it is not super clear. Besides, what is the relationship between WB and IP-WB (see also major comment 4)?
A10: In IP step, the C1-INH protein through 100 μg of IgG-removal serum protein were immunoprecipitated with 2ug of the mouse anti-C1-INH monoclonal antibody coupled to 2 mg of Protein A Sepharose™ CL-4B. In Figure 2A, 2ug of serum protein was loaded to each well from two individual samples (HC and SLE) and immunoprecipitated C1-INH protein was loaded to each well from two pooled samples (20 HCs and 20 SLE) in 8% sodium dodecylsulfate polyacrylamide gel electrophoresis. After gel electrophoresis, a rabbit polyclonal acetylated-lysine antibody was used to detect protein acetylation modification in the Western blot step. The relationship between WB and IP-WB show a flow chart in Supplementary Figure 1.
Q11: Line 127: “Equal amounts”. What do you mean? How it corresponds to Figure 3A?
A11: We loaded 2ug of individual serum protein to each well in 6% sodium dodecylsulfate polyacrylamide gel electrophoresis (Figure 3A, right bottom panel). The red arrow indicates the C1-INH.
Q12: Lines 130-132 and Figure 3B. The combination of sensitivity and specificity you gave is the combination for the best/optimal criterion? The same for lines 162-166.
A12: We could calculate and plot the sensitivities and specificities in different cut points for the levels of C1-INH and Ac-protein adducts. The ROC curve consists of these values for sensitives and 1-specificities. ROC curve analysis provides to decide the optimal cut point for the diagnostic test in medicine. The optimal cut point is selected by the maximum values for specificity and minimum for 1-specificity, simultaneously. So, we always select the cut point in the top-left corner of the ROC. This point is the optimal cut point in our study. The text is “The results obtained from Western blotting indicated that the AUC value was 0.73, sensitivity was 77.5%, and specificity was 52.5% for SLE measurement at an optimal cutoff value of 255624.4 (Figure 3B).” in lines 134-136 on page 6. The text is “The results of the ELISA conducted for determining the serum Ac-protein adduct revealed that the AUC value was 0.67, sensitivity was 70.4%, and specificity was 50.0% for SLE determination at an optimal cutoff value of 0.299 (Supplementary Figure 2, bottom panel).” in lines 167-170 on page 8.
Q13: Lines 185-187. It is a very nice, synthetic, and informative sentence on the relationship between histone acetylation and gene expression. However, please, add 1-2 sentences on other types of histone PT modifications (PMID: 29796022)? Could these other types of PT histone modifications also play a role in SLE and/or RA?
A13: We added the texts “Furthermore, Alaskhar Alhamwe et al. reported that histone modifications including acetylation, phosphorylation, methylation, and ubiquitination may play regulating roles in the development of allergic diseases [33]. In addition to histone acetylation, we investigated other histone modifications that may play a role in the development of SLE. Lupus-derived antibody LG11-2 can react with H2BK14 after apoptosis-induced phosphorylation [32, 34]. Additionally, van Bavel et al. found that apoptosis-induced methylation of H3K27 was targeted by autoantibodies in SLE [14]. Suzuki et al. revealed that antihistone antibodies can bind to ubiquitinated H2A in SLE [35]. Additionally, histone modifications can also affect the development of RA [36]. Lloyd et al. indicated that autoantibody reactions with acetylated histone 2B may play a role in RA pathogenesis [37]. Otherwise, studies on autoantibodies targeted by other histone modifications (including phosphorylation, methylation, and ubiquitination) in RA have not been conducted.” in lines 223-234 on page 10.
Please let me know if any further information is required. Your kind consideration of this submission is highly appreciated.
Yours sincerely,
Ching-Yu Lin, Ph.D.
Professor
School of Medical Laboratory Science and Biotechnology
College of Medical Science and Technology
Taipei Medical University
No. 250, Wuxing Street
Taipei 11031, Taiwan.
Tel.: +886 2 2736 1661x3326;
Fax: +886 2 27324510.
E-mail: cylin@tmu.edu.tw

Reviewer 2 Report
Molecules
Manuscript ID molecules-463336
Low Levels of IgM and IgA Recognizing Acetylated C1-Inhibitor Peptides Are Associated with Systemic Lupus Erythematosus in Taiwanese Women
This paper represents good work in that it is important to determine PTMs on C1-inhibitor (C1-INH) and risk for SLE.
Major comments:
The discussion section was poorly written and it should backup their experimental results.
Minor comments:
Page 1, line 44
Change nuclear acids to nucleic acids
Page 2, line 45-46
The sentence is not clear.
Page number 2, line 47
Make this clear: per 100,000 person-year
Page 2 , line 90
What is the initial mass of 367-LEDMEQALSPSVFKAIMEK-385
and when it is increased to 42.010567 Da, how much difference?
Write this sentence in detail.
Page 2, line 92
MEPFHFKNSVIKVPMMNSK-328
Write the initial mass and increased mass
Page 9,line 185
Change as gene expression, not expressions
Page 9, Line 187:
Change nonhistone to non-histone.
Page 9, line 199
Delete underline at C-terminal
Author Response
Dear Prof. Dr. Paula A. C. Gomes,
We greatly appreciate your interest in our manuscript, “Low Levels of IgM and IgA Recognizing Acetylated C1-Inhibitor Peptides Are Associated with Systemic Lupus Erythematosus in Taiwanese Women” (molecules-463336). We thank the reviewers for their constructive comments and suggestions. We have revised the manuscript accordingly. All the revisions have been highlighted in red in the revised version.
Our responses to the comments and suggestions of reviewer 2 are as follows:
Q1: The discussion section was poorly written and it should backup their experimental results.
A1: We have revised the discussion.
Q2: Page 1, line 44. Change nuclear acids to nucleic acids
A2: We have revised the words “nucleic acids” in line 43 on page 1.
Q3: Page 2, line 45-46
The sentence is not clear
Q3: We have revised the texts “According to the Taiwanese National Health Insurance Research Database, between 2003 and 2008, the average prevalence of SLE in Taiwan was 97.5 new cases (female-to-male ratio, 7.8) per 100,000 persons observed for 1 year; moreover, the highest prevalence in women was observed among those aged 30–39 years, and that in men was observed among those aged 70–79 years. The average SLE incidence rate was 4.87 new cases (female-to-male ratio, 7.0) per 100,000 person-years; the highest incidence rate in women was observed among those aged 40–49 years, and that in men was observed among those aged >70 years. The average standardized mortality rate from SLE was 11.1 new cases (female-to-male ratio, 4.5) per 100,000 person-years.” In lines 44-51 on pages 1-2.
Q4: Page number 2, line 47, Make this clear: per 100,000 person-year
A4: The text “4.87 news cases per 100,000 person-years” imply that there are 4.87 SLE affected in 100,000 people with 1 year follow-up.
Q5: Page 2, line 90. What is the initial mass of 367-LEDMEQALSPSVFKAIMEK-385 and when it is increased to 42.010567 Da, how much difference? Write this sentence in detail
A5: We missed labeling the oxidation of methionine residues in modified peptide (Figure 1B and 1C). We added the text “The initial masses of 367-LEDMEQALSPSVFKAIMEK-385 at charge states of 1 (z = 1) and 3 (z = 3) were 2165.075 and 722.699 Da, respectively. The masses of 367-LEDMEQALSPSVFKAIM*EK-385 charge states of 1 (z = 1) and 3 (z = 3) were 2226.108 and 742.036 Da, respectively (Figure 1B).” in lines 90-93 on page 2.
Q6: Page 2, line 92. 310-MEPFHFKNSVIKVPMMNSK-328 Write the initial mass and increased mass
A6: We missed labeling the oxidation of methionine residues in modified peptide (Figure 1B and 1D). We added the text “The initial masses of 310-MEPFHFKNSVIKVPMMNSK-328 at charge states of 1 (z = 1) and 3 (z = 3) were 2263.132 and 755.384 Da, respectively. The masses of 310-M*EPFHFKNSVIKVPM*MNSK-328 at charge states of 1 (z = 1) and 3 (z = 3) were 2382.174 and 794.058 Da, respectively (Figure 1B).” in lines 97-100 on page 3.
Q7: Page 9, line 185. Change as gene expression, not expressions
A7: We have revised the word “expression” in line 189 on page 9.
Q8: Page 9, Line 187: Change nonhistone to non-histone.
A8: We have revised the word “non-histone” in line 191 and 193 on page 9.
Q9: Page 9, line 199 Delete underline at C-terminal
A9: We have deleted underline at C-terminal in line 203 on page 9.
Please let me know if any further information is required. Your kind consideration of this submission is highly appreciated.
Yours sincerely,
Ching-Yu Lin, Ph.D.
Professor
School of Medical Laboratory Science and Biotechnology
College of Medical Science and Technology
Taipei Medical University
No. 250, Wuxing Street
Taipei 11031, Taiwan.
Tel.: +886 2 2736 1661x3326;
Fax: +886 2 27324510.
E-mail: cylin@tmu.edu.tw

Reviewer 3 Report
The presented manuscript entitled: „Low Levels of IgM and IgA Recognizing Acetylated C1-Inhibitor Peptides Are Associated with Systemic Lupus Erythematosus in Taiwanese Women” describes the identification of new modification of C1-INH and suggests its association with SLE pathogenesis.
Authors have put much effort in the study itself, as well as in the preparation of the manuscript, as it fulfils high international standards. Data are presented clearly and support the presented conclusion. The only flaw of the manuscript are the tables and figures, that in my opinion require enhancement. The font used in the figures and tables is too small. I would also advice to prepare bigger graphs. Moreover, the information about the statistics added to each legend would be the additional value of the manuscript.
Due to my curiosity I would also ask the Authors one question considering samples pooling:
Is there any chance, that due to pooling only 9 randomly chosen samples (it is 1/6 of Your SLE samples), You missed any other interesting, and maybe important, modifications od C1-INH?
The above question does not diminish the value of Your work, as I find it interesting and worth publishing in Molecules. Therefore, after minor revision considering the graphical part of the manuscript, I recommend the presented work for publishing in Molecules.
Author Response
Dear Prof. Dr. Paula A. C. Gomes,
We greatly appreciate your interest in our manuscript, “Low Levels of IgM and IgA Recognizing Acetylated C1-Inhibitor Peptides Are Associated with Systemic Lupus Erythematosus in Taiwanese Women” (molecules-463336). We thank the reviewers for their constructive comments and suggestions. We have revised the manuscript accordingly. All the revisions have been highlighted in red in the revised version.
Our responses to the comments and suggestions of reviewer 3 are as follows:
Q1: Authors have put much effort in the study itself, as well as in the preparation of the manuscript, as it fulfils high international standards. Data are presented clearly and support the presented conclusion. The only flaw of the manuscript are the tables and figures, that in my opinion require enhancement. The font used in the figures and tables is too small. I would also advice to prepare bigger graphs. Moreover, the information about the statistics added to each legend would be the additional value of the manuscript.
A1: We have revised Table and Figures.
Q2: Due to my curiosity I would also ask the Authors one question considering samples pooling: Is there any chance, that due to pooling only 9 randomly chosen samples (it is 1/6 of Your SLE samples), You missed any other interesting, and maybe important, modifications od C1-INH? The above question does not diminish the value of Your work, as I find it interesting and worth publishing in Molecules. Therefore, after minor revision considering the graphical part of the manuscript, I recommend the presented work for publishing in Molecules.
A2: If we have increased sample sizes that may identify more important acetylation modifications in the C1-INH. But, the average coverage of amino acid sequences in the C1-INH was 32% that still limited the possibility.
Please let me know if any further information is required. Your kind consideration of this submission is highly appreciated.
Yours sincerely,
Ching-Yu Lin, Ph.D.
Professor
School of Medical Laboratory Science and Biotechnology
College of Medical Science and Technology
Taipei Medical University
No. 250, Wuxing Street
Taipei 11031, Taiwan.
Tel.: +886 2 2736 1661x3326;
Fax: +886 2 27324510.
E-mail: cylin@tmu.edu.tw

Round 2
Reviewer 1 Report
Thank you very much for addrressing my comments in the professional way. Regarding my previous comments 1 and 2, if it has not been done yet, if possible, please, add the respective senetces to the Discussion.
Author Response
Dear Prof. Dr. Paula A. C. Gomes,
We greatly appreciate your interest in our manuscript, “Low Levels of IgM and IgA Recognizing Acetylated C1-Inhibitor Peptides Are Associated with Systemic Lupus Erythematosus in Taiwanese Women” (molecules-463336). We thank the reviewers for their constructive comments and suggestions. We have revised the manuscript accordingly. All the revisions have been highlighted in red in the revised version.
Our responses to the comments and suggestions of reviewer 1 are as follows:
A: Thank you very much for addrressing my comments in the professional way. Regarding my previous comments 1 and 2, if it has not been done yet, if possible, please, add the respective senetces to the Discussion
Q: We have revised and added the text “Furthermore, autoantibodies against other epitope modifications, including those of N-homocysteinylation, citrullination, MDA, malondialdehyde-acetaldehyde (MAA), carbamylation, acetylation and nitration in non-histone proteins have also been implicated in the etiopathogenesis of SLE and RA [10, 13, 15, 43-47]. Especially, anti-citrullinated and anti-acetylated protein antibody response increase the risk of disease relapse in patients with RA following disease modifying antirheumatic drug (DMARD) treatment [48]. In this study, the levels of anti-C1-INH367-385 Ac peptide antibodies in compared patients with RA with HCs were not significantly different (Figure 4B). Further, we need identify new Ac modifications of serum C1-INH in patients with RA and measure the levels of anti-acetylated CI-INH peptide antibody in patients with RA compared with HCs.“ in Lines 266-274 on page 10-11.
Please let me know if any further information is required. Your kind consideration of this submission is highly appreciated.
Yours sincerely,
Ching-Yu Lin, Ph.D.
Professor
School of Medical Laboratory Science and Biotechnology
College of Medical Science and Technology
Taipei Medical University
No. 250, Wuxing Street
Taipei 11031, Taiwan.
Tel.: +886 2 2736 1661x3326;
Fax: +886 2 27324510.
E-mail: cylin@tmu.edu.tw
